# Stakeholders' perspectives on the status of family planning integration into differentiated antiretroviral therapy service delivery models in Uganda: A qualitative assessment

Joseph K. B. Matovu[1,2]*, Martha Akulume[1], Fredrick Makumbi[1], Elena Lebetkin[3], Rhobbinah Ssempebwa[4], Patrick Komakech[4], Dieudonne Bidashimwa[3], Maria Carrasco[5], Rhoda K. Wanyenze[1]

**1** Makerere University School of Public Health, Kampala, Uganda, **2** Busitema University Faculty of Health Sciences, Mbale, Uganda, **3** Family Health International (FHI360), Washington, D.C., United States of America, **4** United States of America Agency for International Development, Kampala, Uganda, **5** Johns Hopkins Bloomberg School of Public Health, Baltimore, Maryland, United States of America

\* jmatovu@musph.ac.ug

## Abstract

### Introduction

Facility- and community-based differentiated antiretroviral therapy service delivery models (DSDM) for stable patients can offer a convenient platform for integrating self-care-oriented family planning (FP) services into HIV care. However, little evidence exists on the status of FP integration within self-care-oriented DSDM (SC-DSDM). We explored the status of FP integration into SC-DSDM, stakeholders' perspectives about and barriers to integrating FP into SC-DSDM and suggestions for improving FP integration into SC-DSDM.

### Methods

This qualitative study was conducted in 18 purposely-selected health facilities in 17 districts across four high HIV-prevalence regions between September and October 2022. We conducted 36 in-depth interviews with women living with HIV (i.e., clients), receiving ARV drug refills through SC-DSDM, and 47 key informant interviews with selected stakeholders including healthcare providers, district health managers, implementing partner representatives and policymakers. Data were collected on the different forms of FP integration into SC-DSDM; perspectives on integrating FP into SC-DSDM, and barriers to and suggestions for improving FP integration into SC-DSDM. Data were analyzed following a thematic framework approach.

**Data availability statement:** Data cannot be shared publicly because of the qualitative nature of the data which cannot be completely de-anonymized. Data are available from the School of Public Health Research and Ethics Committee (sphrecadmin@musph.ac.ug) for researchers who meet the criteria for access to confidential data.

**Funding:** This manuscript is made possible by the support of the American People through the United States Agency for International Development (USAID) for the Research for Scalable Solutions (R4S) Project under Cooperative Agreement #7200AA19CA00041, awarded to Family Health International (FHI) 360. The R4S Project is a global project that was implemented by a consortium led by FHI 360 in partnership with Evidence for Sustainable Human Development Systems in Africa (EVIHDAF), Makerere University School of Public Health (MakSPH), Population Services International (PSI), and Save the Children (STC). The findings reported in this paper originate from a subgrant awarded to MakSPH (PI: RKW) from FHI 360 to implement a variety of R4S Project activities in Uganda. The contents of this manuscript are the sole responsibility of FHI 360 and do not necessarily reflect the views of USAID or the United States Government.

**Competing interests:** The authors have declared that no competing interests exist.

## Results

We found two forms of FP integration into SC-DSDM: a) one-stop center (in which ART and FP services were provided at the same service delivery point) and b) collaboration/referral to another service delivery point. Only four health facilities offered ART and FP services through the one-stop center; the rest of the health facilities referred clients to the maternal and child health/FP clinic or to other health facilities. All categories of stakeholders agreed that the one-stop center is more convenient and less time-consuming since referral to another service delivery point can increase patient waiting time or result in multiple clinic visits if ART and FP services are offered on separate days. Staff shortages, stock-outs of short-term FP supplies, shortage of adequate office space and lack of integrated registers continue to hamper effective integration of FP into SC-DSDM.

## Conclusion

Despite the potential benefits of FP-HIV integration, FP and ART services continue to be offered as stand-alone programs with limited FP integration into SC-DSDM. These findings call for policy guidance from the Ministry of Health in integrating FP into SC-DSDM in Uganda.

## Introduction

Differentiated ART service delivery (DSD), previously referred to as 'differentiated HIV care', is a client-centered approach that simplifies and adapts HIV services in ways that not only meet the HIV clients' needs but also reduce unnecessary burdens on the health care system [1]. Differentiated ART service delivery has been scaled up in many countries in recent years [2] with the goal of improving efficiencies in HIV care through reduced patients' waiting time, reduced patients' visits to the health facilities, and reduced health providers' workload due to seeing patients less frequently through multi-month dispensing of ARV drugs [3]. As a result, most people living with HIV are now seen by clinicians or visit health facilities less frequently [4–6]. Concurrently, there is a renewed emphasis on ensuring that all women living with HIV (WLHIV) have access to other non-HIV services, including family planning (FP), as part of their HIV care and as an essential pillar in the prevention of mother-to-child transmission of HIV [7–9]. This renewed emphasis on FP as an essential part of HIV care follows closely on recent calls for more inclusive and patient-centered care for people living with HIV [10,11]. However, WLHIV who visit health facilities to collect their ART do not fully benefit from patient-centered care if they have to return to the same health facility more frequently to receive other non-HIV services, including FP. Clients in DSD models should be supported to use contraceptive methods of their choice and, where ongoing commodities are required, be able to obtain these through their differentiated ART service delivery models (DSDM) [1].

One way in which WLHIV can be supported to access the contraceptive methods that they need when they come for their ARV drug refills is to streamline the

schedules for contraception and ART resupplies in such a way that the client can be provided with FP supplies of the same duration as their ART refills [12]. Under an ideal integrated FP and HIV service delivery mechanism, FP and ART care should be provided at the same time, in the same place and, where possible, by the same provider [13]. This can be achieved through different approaches, including use of multi-skilled HIV and FP providers and a one-stop service center within the ART clinic site (or at the community drug distribution point) offering HIV and FP services under one roof [14]. In Tanzania, for example, women in DSDM have access to a full range of contraception options; ART and FP are both offered within the same care and treatment centers [15]. This alignment ensures uninterrupted supply of contraceptives and ART while minimizing visits to the health facility [1]. Similar arrangements can be made for community-based ART delivery approaches, including sending out multi-skilled HIV and FP providers to manage ART refills and FP supplies at the community outreach service delivery points [16]. Similarly, for community group models, provision of FP can be aligned with the scheduled FP refill of the group member who goes to the health facility (or community drug distribution point) to pick ART for the other members [1].

In 2016, Uganda made DSDM a priority in the national HIV prevention, care and treatment guidelines so that HIV service delivery could be more client-centered and scarce resources could be refocused to those most in need [17]. In 2017, the Ugandan Ministry of Health developed the national policy guidelines and service standards for sexual and reproductive health services and rights (SRHR) [18] that called for integrating STI/HIV and AIDS within SRHR services. However, these guidelines did not include details on how STI/HIV and AIDS integration into SRHR services would be implemented. In tandem with this observation, the Ministry of Health, in 2017, developed DSDM implementation guidelines that were more aligned to HIV service delivery [19]. These guidelines were expected to provide guidance on the integration of FP into the differentiated ART service delivery models. However, until this assessment, there was no formal evaluation to determine if FP integration was well streamlined within the DSDM guidelines, and if so, whether integration of FP into the DSDM was being implemented based on these guidelines.

Differentiated ART service delivery is now the primary mode of providing ART refills to people living with HIV (PLHIV) in Uganda with the majority (65.6%) of the patients categorized as stable clients (or clients on less intensive models) while 34.4% are unstable patients/clients on more intensive models [20]. Stable patients are those with near-perfect ART adherence and who have achieved virological suppression while 'unstable patients' are those that still have ART adherence challenges and are still virally non-suppressed [19,20]. In Uganda, DSD models for stable patients include a) Fast-track drug refills (FTDR) in which HIV-positive stable patients in care for six or more months, who are TB-free, pick their drugs directly from the health facility's dispensary or ART clinic once every three months; b) community drug distribution point (CDDP) in which healthcare providers provide ART services to stable patients in a given locality (e.g., at lower-level health centers, private pharmacy or other locations) in the community on a rotational basis, once every three months; and c) Community client-led ART delivery (CCLAD) in which stable patients (who have been on HIV treatment for six or more months) are arranged into groups of 3–6 patients, one of which goes to the facility of their ART registration to pick ARV drug refills on behalf of the other members on a rotational basis (once every three months). DSD models for unstable clients include: a) Facility-based individual management and b) Facility-based groups [19,20]. This paper focuses exclusively on DSD models for stable clients.

Given that the Uganda Government has prioritized self-care as part of its strategy to reach universal health care coverage [21], this study focused on three differentiated ART service delivery models for stable patients (fast-track drug refill, community drug distribution point, and community client-led ART delivery), hereafter referred to as 'self-care-oriented DSDM' (SC-DSDM), due to their potential increased efficiency within the HIV care system while enabling clients to receive self-care-oriented FP services. We defined self-care-oriented FP services as FP services that WLHIV, enrolled in SC-DSDM, can access and use with minimal or no provider interaction, and for which they can receive FP supplies or services of the same duration, if needed, as their ART refills. However, while integrating FP into the DSDM in general and the three SC-DSD models in particular would improve access to FP services and methods and reduce the frequency of client visits

to the health facility, by the time of this assessment, no formal evaluation had been conducted to document the status of FP integration into the self-care-oriented DSD models. This assessment aimed to explore: a) the status of FP integration into the three SC-DSDM, b) client, health provider, implementing partner and policymaker perspectives about and barriers to integrating FP integration into the SC-DSDM, c) whether or not FP-HIV integration was clearly defined (in terms of how it should be done, when, and by whom) within the existing DSDM guidelines, and d) suggestions on how to improve FP-HIV integration within the DSDM framework in Uganda.

## Methods

### Study site

This qualitative study was conducted in 17 districts in four purposely-selected geographical regions with the highest HIV prevalence in Uganda, based on the findings from the 2020–21 Uganda Population-based HIV Impact Assessment survey (hereafter referred to as 'UPHIA 2020-21') [22]. UPHIA 2020–21 was a nationwide survey conducted to provide estimates of HIV prevalence, viral load suppression, and other important HIV/AIDS program indicators [22]. The selected regions included: 1) *Central-1* (HIV prevalence – 8.1%); 2) *Mid-North* (HIV prevalence – 7.6%); 3) *South-western* (HIV prevalence – 6.3%) and 4) *Central-2* (HIV prevalence – 6.2%).

### Study design and population

This was a qualitative study conducted among WLHIV who had ever used a FP method since joining a self-care-oriented DSDM; SC-DSDM implementers at health facility level (ART clinic in-charge, Facility In-charge, and Maternal and Child Health [MCH]/FP clinic in-charge); district health officials (District Health Officers [DHO], Assistant DHO in-charge of MCH, and/or district FP focal person); staff from implementing partner (IP) organizations that support health services in each region, and policymakers. Policymakers included national-level staffs who were involved in the design and rollout of the DSDM guidelines countrywide.

### Selection of health facilities and participants

**Selection of health facilities.** In Uganda, government-owned/public health facilities are categorized as health center level-II (HC-II) which serves an estimated population of 5,000 people; health center level-III (HC-III) which serves an estimated population of 20,000 people; health center level-IV (HC-IV) which serves an estimated population of 100,000 people; district/general hospital which serves an estimated population of 500,000 people; regional hospital which serves an estimated population of 1,000,000 people; regional referral hospital which serves an estimated population of 2,000,000 people, and national referral hospital which serves an estimated population of 10,000,000 people [23]. Only health facilities at the level of HC-III or higher (or as otherwise classified by the Ministry of Health, if they are private health facilities) are accredited to offer ART in Uganda.

Using data from the Ministry of Health's District Health Information System Software-version 2 (DHIS-2), we documented the names of all the ART-accredited health facilities in each region and categorized them into high- and low-volume health facilities based on the number of people living with HIV (PLHIV) enrolled in HIV care as of June 2022. Health facilities with less than 500 PLHIV in care at the time were categorized as low-volume health facilities while those with 500 PLHIV or more in care were considered to be high-volume facilities. The cut-off (i.e., 500 PLHIV) for categorizing a health facility as 'low-' or 'high-volume' was determined arbitrarily, assuming that health facilities with 500 PLHIV or higher were more likely to have patients registered in the different self-care-oriented DSD models of interest. We purposively selected 18 health facilities across the four regions, including 14 government-owned/public health facilities and four private not-for-profit health facilities.

Initially, all the six regional referral hospitals that exist within the four regions were selected (i.e., Mbarara and Kabale Regional Referral Hospitals in the *Southwestern region*; Masaka Regional Referral Hospital in *Central-1 region*; Mubende

Regional Referral Hospital in *Central-2 region*, and Lira and Gulu Regional Referral Hospitals in the *Mid-Northern region*). These hospitals were included because they have a high patient load and were, therefore, presumed to implement several DSDM, including the self-care-oriented models for stable clients (i.e., facility- and community-based models). After selecting the regional referral hospitals, we purposely selected three lower-level health facilities within each region for a total of 12 health facilities. The 12 health facilities included three hospitals (two district/general hospitals and one private hospital), four government-owned/public HC-IVs, three HC-IIIs (two government-owned/public and one private health facility), and two special HIV clinics run by a national-level non-government organization. We included the two special HIV clinics because the organization that runs them has a long-standing experience in the implementation of community DSD models. Of the 18 health facilities, five were selected from the Mid-Northern region; five from the South-Western region, four from Central-1 region and four from the Central-2 region.

**Selection of participants.** Eighty-three participants were interviewed for this study, including 36 WLHIV receiving HIV care who were receiving their ARV drug refills through the three SC-DSDM (also referred to as 'clients' elsewhere in this paper); 35 healthcare providers working at the selected 18 health facilities; four (4) district health officials; five (5) IP staff representatives, and three (3) policymakers. To participate in the study, clients had to be stable patients, enrolled into one of the three (3) self-care-oriented DSDM, and have used a FP method since joining DSDM.

At each health facility, we worked with the in-charge of the ART clinic to identify eligible WLHIV, stratified by the type of SC-DSDM they were enrolled in. The identified clients were recorded on a pre-printed participant log. The ART in-charges made initial contact with the clients (via phone contact or through a provider or linkage facilitator, as appropriate) to inform them about the study and inquire about their interest in participating in the study. While the use of the ART clinic in-charges to contact WLHIV might have influenced their participation in the study in some way, women had the right to decline participation without this (i.e., declining to participate in the study) affecting their access to routine HIV care at the same health facility. Interested WLHIV were connected to the data collector via phone (or through a provider or linkage facilitator, as appropriate) to provide them with more information about the study and conduct initial screening on the phone to confirm compliance with the study eligibility criteria. To confirm eligibility, stable clients (as designated by the ART clinic) were asked about their DSD model of enrolment and if they had ever used a FP method since they joined the DSDM. Stable clients, enrolled in any of the three SC-DSDMs and who had used a FP method since they joined the DSDM, were eligible for this study. Clients who were confirmed to be eligible were invited for an interview at the health facility. We sought informed consent from all the clients prior to their participation in the study.

Regarding healthcare providers, we selected either the health facility in-charge, ART in-charge, or the MCH in-charge/FP focal person. The study team contacted the participants by phone or in person at the health facility to inform them about the study and confirm their availability. For district health staff, we obtained the names and telephone contacts of the DHOs (who are the substantive heads of health programs in their districts) from the Ministry of Health and contacted them initially via the phone. If they were willing to be interviewed, we set an appointment with them either at the district headquarters or at any other venue that they considered appropriate. Through the DHOs, we obtained the names and telephone contacts of the other district health staffs (Assistant DHO in charge of MCH or District FP Focal Person) who were also contacted and an appointment was set with them if they were willing to be interviewed. We used the same strategy while approaching the implementing partner organizations in each region. Implementing partner organizations were asked to recommend staffs who were invited for interview at an appropriate time and venue. For policymakers, we set appointments for physical interviews through telephone calls to the respective policymakers.

## Data collection procedures and methods

Data were collected during the months of September and October 2022 by a team of thirteen trained data collectors. In-depth interviews were conducted with WLHIV while key informant interviews were conducted with health facility staff, staff representatives from IP organizations, district health officials and policymakers using unstructured interview guides.

Clients were asked questions such as: "*Have you used family planning in the past?", "What places have you gotten [FP] methods from in the past?*", and "*Please think back to the past year, have you gotten your [FP] method at the same time that you got your ARV*?" (See **Supplementary File 1** in S1 File for details). On the other hand, key informant interviews were asked questions like: "*Could you please describe how integration of FP into DSDM is meant to take place in the health facilities?*", and "*What guidelines were followed during the process of integrating FP into DSDM?*" (See **Supplementary Files 2–4 in** S1 File for details). The interviews with clients were conducted in the local language that they were comfortable with while key informant interviews were conducted in English. All the interviews were audio-recorded with permission from the participants. Interviews lasted between 45–90 minutes. Prior to data collection, the data collection team underwent fieldwork preparation training for three days. As part of the training, the data collection team pre-tested the data collection tools in a health facility in a non-study district to test data collection procedures, ensure clear wording of the data collection tools, and evaluate the readiness of the data collection team. Areas of weakness, identified during the pre-testing exercise, were corrected prior to fieldwork.

### Data analysis

The audio-recordings were transcribed verbatim (in the same way they were conducted in the local language) and later translated into English for data analysis. An independent researcher, conversant with the local language as well as the English language, reviewed the transcripts along with the recordings for quality assurance. The transcripts were then used to conduct a Microsoft Excel-based, thematic matrix analysis. Data analysis followed the steps of the thematic framework approach, as suggested by Braun and Clarke [24]. Initially, AM read through twenty transcripts to get familiar with the data and generated a tentative list of initial codes. Using the tentative list of codes, AM searched for deductive themes of inquiry including forms of FP integration into the SC-DSDM, FP-HIV integration into the community-based DSD models, alignment of FP-HIV integration to the DSDM guidelines, perceptions about integrating FP into the SC-DSDM, barriers/challenges to FP-HIV integration in the SC-DSDM and suggestions for improving the integration of FP into the SC-DSDM. AM and JKBM reviewed the codes for each theme to check for "referential adequacy" by comparing the codes with the raw data to ensure that they accurately reflected the meanings intended by the participants [25]. Inadequacies in the codes were rectified through discarding those that were inconsistent with the themes or through adding new ones. AM and JKBM met to review and generate the final list of codes. Using the final list of codes, AM used *Atlas.ti* qualitative data analysis software (version 9.0) to check the remaining transcripts for relevant quotations that pertained to each *a priori* theme and copied and pasted them into the Excel matrix. JKBM, AM and RKW reviewed the matrix to identify rich "textual" quotations that were used in supporting the key findings presented in this paper. We followed the consolidated criteria for reporting qualitative studies (COREQ) in reporting the findings [26].

### Ethical considerations

This study was approved by the Makerere University School of Public Research and Ethics Committee (protocol number SPH-2021–185) and registered by the Uganda National Council of Science and Technology (HS2026ES). Permission was also sought from the relevant district officials and health facility in charges. All participants provided written informed consent prior to study participation and consent to an audio-recorded interview.

## Results

### Characteristics of the study participants

Table 1 shows the number and categories of participants included in this study. The distribution of WLHIV mimics the general distribution of the patients across the different DSDM, with the highest number of participants drawn from FTDR and the lowest number from CCLAD. Generally, there were few or no WLHIV enrolled in the CCLAD model across facilities.

**Table 1. Number and category of participants included in the study.**

| Participants | Category of participant | Number of participants |
|---|---|---|
| Women living with HIV | Fast-track drug refill (FTDR) client | 19 |
| | Community drug distribution point (CDDP) client | 11 |
| | Community client-led ART delivery (CCLAD) client | 6 |
| | **Sub-Total** | **36** |
| Health providers | Health facility In-charge | 13 |
| | ART In-charge | 14 |
| | Maternal and child health In-charge | 8 |
| | **Sub-Total** | **35** |
| Policy-makers, implementing partner representatives and district health officials | Policymakers | 3 |
| | Implementing partners | 5 |
| | District health officials | 4 |
| | **Sub-Total** | **12** |
| **Total** | | **83** |

### Forms of FP integration into the SC-DSDM

At the time we conducted the study, FP-HIV integration in general was already taking place. However, we did not know if and how FP integration into the DSDM was taking place. To understand if and how FP integration into the DSDM was taking place, we asked women to tell us about where they access their FP services and methods when they come for their ART refills. In response, a few of the women indicated that they received their FP methods from the same service delivery point where they obtain their ART refills while the majority indicated that they are referred to the maternal and child health [MCH] clinic/FP clinic. Thus, FP integration was deemed to take two different forms: a) integration through the one-stop center and b) integration through referral to another service delivery point. The two forms of FP-HIV integration are summarized below.

**a)** ***FP-HIV integration through the one-stop center (i.e., combined service provision).*** All the 18 health facilities offered FP information and counselling to WLHIV in care through the three self-care-oriented DSD models (i.e., FTDR, CDDP, and CCLAD). However, only four health facilities offered FP services (including FP methods) and ART refills to WLHIV through the one-stop center. The four health facilities included one (1) government-owned/public Regional Referral Hospital, one (1) private-for-profit hospital, and two (2) non-governmental organization (NGO) special HIV clinics. At these health facilities, clients received both ART and FP services at the same place due to the presence of space to offer both services and a team of "knowledgeable" (multi-trained FP-HIV) health workers, as the following quotations illustrate:

> *When I come here [at the CDDP] to get my ARVs, there is … a place where we get our ARVs from; that is where we go… After giving you your ARVs, then they give you the injection … at the same point because we have a room… in that very room, I leave with my packed ARVs; they have [also] given me my FP injection (**CDDP client, Central-2 Region**)*

> *"Being a regional referral hospital, we have most of the FP equipment … family planning methods, the knowledgeable health workers … stationed in the ART clinic to provide the family planning services to those patients" (**Healthcare provider, South-Western Region**)*

Although some health provider in-charges indicated that FP had been integrated into the SC-DSDM, there were cases where client interviews reviewed opposite information, with some clients reporting that they were instead referred to the MCH/FP clinic for FP services. However, some health provider in-charges acknowledged that they had tried to integrate

FP services into the SC-DSDM but faced challenges in maintaining regular stocks of FP supplies. In such facilities, the offer of FP through the SC-DSDM had ceased to be provided.

*"We used to go with FP supplies (condoms, injectables and pills) to community outreaches but this stopped last year due to la6ck of supplies (**Healthcare provider, Central-1 Region**)*

**b)  *FP-HIV integration through referral (intra- or inter-facility referral).***  Of the 18 surveyed health facilities, 13 offered FP services (including methods) through referral to the MCH/FP clinic on the same compound as the ART clinic while one (1) health facility (a faith-based health facility) referred clients interested in receiving FP methods to other health facilities. In health facilities that refer to the MCH/FP clinic, some healthcare staff reported that WLHIV are escorted to the MCH/FP clinic as part of the referral process, as illustrated below.

*".... what we do, we health educate and select those who are willing to get FP services every day, for those who are willing, we get somebody to escort them to MCH, we make sure they are worked on very fast so that [they] can go back to the clinic to get their drugs."* (**ART provider in-charge, Southwestern Region**)

However, in general, most WLHIV noted that when they were referred from the ART service delivery point to the MCH/FP clinic, they were not escorted to the MCH/FP clinic and had to join the queue with the other patients who were waiting to be served at that point. This form of non-coordinated, intra-facility referral increased the time that WLHIV who had come to pick their ARV refills but were also interested in getting FP methods spent in the health facility.

### FP integration into the community-based DSD models

Integrating FP services into the community-based DSD models can provide a useful opportunity for leveraging the ART platform to offer FP services to WLHIV at the community level. This can help to reduce the need for WLHIV to come to the health facilities for FP services when they can receive them as part of community-based DSD models. In response to the question as to whether WLHIV, who receive their ARV drug refills through CDDP or CCLAD, receive FP services at the same time as their ARV drugs, most women indicated that they have to make a separate visit to the health facility for FP services. This finding was corroborated by some of the healthcare providers who confirmed that health outreach teams (which deliver ARV drugs to the community distribution points) usually do not carry FP methods along with the ARVs, with the exception of condoms.

*"Clients do not get FP supplies at the CDDP – these points act only as ART pick-up points. No FP services are provided there"* (**ART provider in-charge, Central-1 Region**)

During interviews with the healthcare providers, we found that there was very limited integration of FP into the community DSD models largely because the ART team prioritized ART to FP provision at the community outreach sites, and usually did not carry a mix of short-term FP supplies with them to those sites, except condoms. The only exception was the two NGO special clinics, which provided some FP methods through the CCLAD model. However, even then, our team was not able to document how FP services are provided to clients in the CCLAD model given that this model is organized to have only one person come for ART refills for the other members, usually once every three months. Otherwise, in the rest of the health facilities, integration of FP into the CCLAD model seemed not to be taking place:

*"… but in that [CCLAD] group, in case there is a client who is on injectable, we are not able to dispense to the [CCLAD] leader. We shall link that client to the nearby health facility to receive the method. But if the client in that group is able to come to the facility, then we shall be able to provide the family planning method she wants"* (**Health care provider, South-Western Region**)

Besides the limited FP integration into the community DSD models, we found that the implementation of the community DSD models had stalled in some geographical regions due to limited facilitation for community outreach activities. For instance, in the Central-2 region, only two of the four health facilities surveyed had active patients in the CCLAD model. Likewise, in the South-West region, only three of the five health facilities surveyed had active clients in the CDDP or CCLAD model.

## Alignment of FP integration with the DSDM Guidelines

As noted earlier, one of the study objectives was to explore whether FP integration into the DSDM was well streamlined into the DSDM guidelines, and if so, whether integration of FP into the SC-DSDM was being implemented based on the DSDM guidelines. Our findings show that FP integration into the DSDM was not comprehensively discussed during the design of the guidelines or considered during the training of health providers prior to the roll-out of the DSDM guidelines. Specifically, the training of health providers on the implementation of the DSDM guidelines focused exclusively on improving efficiency in ART provision with no specific reference to FP integration:

> *"I can ably say that in my own assessment... FP was not given that due attention [during the training]. We were focusing on the general principles of DSDM so when we reached on what services we were supposed to offer [under DSDM], I think our focus was mainly on ARVs, isoniazid prophylaxis, anti-TBs"* **(Implementing partner, South-Western Region)**

> *"I participated in rolling out the [DSDM], and training of health workers on the different models but am not aware there was a specific training on FP so it might not have been emphasized"* **(District Health Official, South-Western Region)**

In interviews with the policymakers who were involved in the design and rollout of the DSDM guidelines at national level, we were informed that since there were already FP services within the health facilities (either through the MCH or FP clinics); it was presumed that a different FP service delivery point at the ART clinic was not necessary. Instead, the policymakers presumed that clients would access FP services through intra-facility referrals to the MCH/FP clinic.

> *"I think we did not think so much about family planning at that time [of developing the guidelines]. We concentrated more on HIV, access to HIV services. We had not concentrated on integration [of family planning], the only thing we integrated early were those things which are closely related to HIV like TB"* **(Policymaker, National level)**

## Perceptions about FP integration into the DSDM

When we interviewed clients regarding their perspectives on integrating FP into the DSDM, all of them agreed that integrating FP into the DSDM as part of the HIV care clinic is convenient and less time-consuming, when compared to being referred to the MCH clinic. A client from a facility where integration is taking place noted:

> *"… when you reach there at the facility, you can easily get all these [ART and FP] services at once from the same place, you receive the ARVs and […] the family planning injection [at the same time]"* **(FTDR client, South-Western Region)**

SC-DSDM clients reported that it is inconveniencing to receive FP services from the MCH clinic because they have to line-up again (and join long lines before being served) and that, at times, FP and ART services run on different days, meaning that when they get one service (e.g., ART refills) on any typical day, they would have to come back to the facility on another day to receive another service (e.g., FP).

*"… sometimes there are many people at family planning [unit] so I come early and I end up leaving late. I would wish that [the services were on] the same day. That's the only challenge I have found, time. Because I come around 9am and leave around 2pm"* (**FTDR client, South-Western Region**)

Providers and implementing partners thought that FP-HIV integration into SC-DSDM would reduce waiting time at the health facility and reduce unmet need among HIV clients.

*"For me I say [FP-HIV integration into SC-DSDM] is very good. Because it reduces the time clients take within the hospital. The client will come, get all the services in one place and go, instead of referring her to the MCH, where she finds the line is too long, and she waits for 4-5 hours yet she stays far"* (**Healthcare provider, Central-2 Region**)

### Barriers/challenges to FP integration into the DSDM

Barriers/challenges to FP integration into the DSDM were explored from the providers' and implementing partners' point of view. The most common barriers, cited by both providers and implementers, were: staff shortages, stock-outs of short-term FP supplies, shortage of office space, difficulty synchronizing FP and HIV service return-time, increased workload, and lack of integrated registers, as indicated in the following sub-sections.

a) *Staff shortages*. Across health facilities, health provider in-charges reported that they had few health providers allocated to the HIV clinic and that given the heavy client load in the HIV clinic, most health providers focus on ART service provision at the expense of FP services. In one health facility which had a designated health worker to serve WLHIV interested in FP services when they come to the ART clinic, we were informed that the said health worker had been transferred to another health facility without any replacement plans in place.

*There used to be a trained midwife who offered FP in the ART clinic. However, she is on maternity leave and on transfer to Soroti Regional Referral Hospital. There is no plan for a replacement* (**ART health provider in-charge, Mid-Northern Region**)

b) *Stock-outs of short-term FP commodities*. Persistent stock-outs of FP supplies were reported across all health facilities to the extent that in at least one health facility, providers had resorted to encouraging clients to use long-term reversible methods that were more commonly available than short-term FP methods that were always out of stock. This has implications for the available FP methods-mix that WLHIV can choose from, depending on their preferences. At another health facility, health providers reported that short-term FP supplies run out so fast yet there is no replenishment plan in place.

*[We used to have] someone who would offer Sayana press to [WLHIV] in the HIV clinic but supplies ran out in only two months and there has not been any replenishment* (**ART health provider in-charge, Central-1 Region**)

*"Another challenge [to integration] is that FP commodities are not there. If they would be providing us with a constant [supply of] these short-term commodities and there are no stock-outs, then our service delivery will be perfect."* (**Healthcare provider, Mid-Northern Region**)

c) *Shortage of adequate office space*. Health providers reported that one of the challenges they have faced regarding integrating FP into the SC-DSDM is the issue of space. Health providers reported that the limited HIV clinic space was not sufficient to enable them to provide some FP methods, especially those that would require client privacy.

*"FP-HIV integration into SC-DSDM is okay but the challenge we have now is space, because initiating any mother or any client on family planning, we first need to examine and now at the ART clinic, space for examination of mothers it is not there"* (**Healthcare provider, South-western Region**)

*"… I would say our space [is a challenge]. By the time we made that construction in ART clinic, family planning program was not there but now we have to squeeze in the limited space we are having, it is where we do the screening for cervical cancer, then again it is where we have FP, it is the same place"* (**Healthcare provider, Mid-Northern Region**)

**d) Difficulty in synchronizing FP and HIV return-time and duration of supplies.** Another important challenge was that FP and ART do not have synchronized schedules. For this reason, while a woman may come for their ART drug refill on a particular day, it was likely that FP services were provided on another clinic day which required them to come back at another time. This lack of synchronization of FP and ART schedules made women to resort to obtaining their FP supplies elsewhere rather than return to the same health facility on another day to receive their FP supplies. In addition, while women would be given ART drug refills for six months, there was no indication that women who used contraceptive pills, for instance, were given supplies to last them until their next ART refill visit date.

*"… [FP-HIV integration into SC-DSDM] is a good thing but the implementation is not as smooth … I just want to give an example of someone in [in the] fast-track model; right. This mother is going to…. take ARVs for six months and maybe on that day she gets her family planning shot, in those six months she will be expected to have gotten another family planning shot"* (**Implementing partner, Mid-Northern Region**)

**e) Use of different registers for FP and HIV data capture.** At health facility, FP and HIV services are captured in different registers. Since the FP register is usually located at the MCH/FP side of the health facility, providing FP services to women receiving their drug refills through the SC-DSDM would require that the ART provider go to the MCH/FP clinic to complete the FP register; something that most providers find burdensome and time consuming. For instance, in the health facilities where FP services were provided at the same time as the ART drug refills, some ART providers use a paper-log to capture the details about their ART client who had received FP services, and then use the paper-log to complete the FP register at another time. Given the staff shortages in the ART clinics and the burden that would come with providing FP services through the SC-DSDM, most of the health workers preferred to refer the women to the MCH/FP clinic rather than offer FP services at the same ART service delivery point.

*"The issue is about the documentation and even the accountability of the commodities. We are not supposed to be having 2 data tools [registers] within the facility, it is supposed to be one. So, it is better to offer it [FP] through MCH"* (**Healthcare provider, Central-1 Region**)

What this healthcare provider was trying to emphasize was the fact that health facilities are not expected to maintain two registers for the same service (e.g., have a FP register at the ART clinic and a different copy at the MCH/FP clinic), it is better to have the FP register at the MCH/FP clinic which traditionally provides FP and other maternal-related services in health facility settings.

**f) Inadequate facilitation for outreach activities.** Inadequate financial facilitation and/or lack of transport to go for community outreaches were cited as one of the main challenges faced by health workers as they integrated FP into community-based DSDM. Some health providers reported that while they had started to operate the community-based models, they had to close them due to lack of funding or client instability.

*"Those ones [the CCLAD] we tried at the beginning when we were starting the community delivery model but it didn't pick up … They [clients] were shifting from [one] place to the other, others went to school"* (**Healthcare provider, South-Western Region**)

*"It [the CDDP arrangement] didn't go on because of facilitation. There's a challenge of transport. Even when these health workers go to the field, at least they need like a bottle of water, but now the finances are limited. So, it couldn't go on"* (**Healthcare provider, Central-2 Region**)

**Suggestions on how to improve FP integration into the SC-DSDM**

To address the above-mentioned barriers/challenges to FP integration into the SC-DSDM, implementing partners, health-care providers and district health managers suggested a need to: a) build the capacity of existing HIV staff in the provision of FP services; b) recruit additional health workers to deal with the persistent staff shortages; c) provide integration guidelines to the health facilities; and d) merge reporting tools. These points are elaborated below.

a) ***Build the capacity of existing ART staff to offer FP services*.** To improve the integration of FP into the SC-DSDM, most health care providers suggested that HIV providers should be continuously trained in the provision of FP services. For the CCLAD model, the health providers recommended that expert clients should be trained to provide FP services to fellow clients in the community.

*"Mainly it could be better if all the people that are attached to the ART clinic are trained to offer all the family planning services offered to the clients. If that is done, it means that these clients will be receiving services from one point other than being referred to the MCH clinic"* (**Healthcare provider, Central-1 Region**)

*"The other thing I have seen is that for the CCLAD their expert client requires training like what we did for the Village Health Teams (VHT) so that when he takes for them family planning methods and (he/she) is asked questions and he/ she can answer. The VHT members were trained and showed how to inject for family planning therefore the expert client needs to be trained especially on how to inject for family planning because an injection must not just be given by one who is not trained"* (**Healthcare provider, Central-2 Region**)

b) ***Recruit additional health workers to minimize staff shortages*.** In addition to building the capacity of available staff through training, some of the health care provider in-charges recommended the need to recruit more health workers to address staff shortages and reduce the workload created by the need to provide FP at the same service delivery point on top of HIV services.

*"Maybe they [the Ministry of Health] should […] recruit more health workers to deal with the large clientele. I know because the line is long in the [HIV] clinic because of the number of clients they receive and now the integration, so they have to make sure that they bring more health workers to give the service [integrated services]. That is what I think, because there are delays to the clients as they must wait for the available health workers to offer the different integrated services to them"* (**Healthcare provider, South-western Region**)

c) ***Provide clear guidelines on FP integration into the SC-DSDM to health providers*.** A few health workers acknowledged that there were no clear guidelines on how they should integrate FP into the SC-DSDM at the health facilities. They therefore recommended the need to provide clear guidelines to guide health workers in the provision of FP services integrated into the SC-DSDM. The availability of these guidelines should help to address some of the challenges already mentioned in this paper, including the need for synchronized FP/HIV return times and duration of FP supplies; how WLHIV receiving FP services at a different service delivery point (other than the HIV clinic or CDDP) should be handled when they come for FP services; and what data should be collected, when, and using what data capture tools when WLHIV under the SC-DSDM receive FP services:

*"We don't have a clear guideline of the integration. There is no document you can look at to know how we integrate this; there is no such document, there are no standard operating procedures to refer to. We need supporting guidelines*

*to help the health worker to routinely refer to, to make the integration better"* (**Healthcare provider, Mid-Northern Region**)

**d) *Merge FP and HIV reporting tools*.** Some participants recommended the need to merge the reporting tools to reduce the burden of having to fill in two different (FP and HIV) registers where a person received FP and HIV services.

*"… We have an independent register for family planning; can we have the same dispensing log to capture ART commodities and family planning commodities? Because you find a health worker after writing a huge book for ART, you close the chapter; open another huge book for family planning. Let them have one register for that! Yeah. That should be changed completely because most problems are in the work you have to do when the patient has left. You find people struggling with books. So, they should look into that"* (**Healthcare provider, Central-2 Region**)

## Discussion

This study found that all the 18 facilities sampled offered FP information/ counseling through the three self-care-oriented DSDM but only four facilities provided FP methods in the same ART clinic or in the same community outreach visit. In general, we found that: a) most of the clients receiving their ARV drug refills through the facility-based DSD models have to access FP methods through referral to the MCH/FP clinic, located outside the ART service delivery point while b) most of the clients receiving their RV drug refills through the community-based DSD models have to make a separate physical visit to the MCH/FP clinic, usually on another day, to obtain their FP supplies.

The finding that only four out 18 health facilities had FP integrated into the SC-DSDM is frustrating given current efforts to support the integration of FP into non-HIV services in Uganda [27]. However, these findings are not unusual [28–30]. In an evaluation of FP integration into the DSDM in Kenya in 2020, the Elizabeth Glaser Pediatric Foundation found that FP services were not integrated into the comprehensive HIV care clinics in Kenya but required referral to the MCH unit. No FP services were integrated into ART refill visits, either in facility or community settings; as a result, all pre-packed ARVs for distribution in facility or community models did not include contraceptive commodities. Besides, ART and FP refills and visits were not formally aligned [30]. However, the successful integration of FP into services offered at HIV care clinics in other African settings has also been documented. In Tanzania, for example, contraceptive care was provided at the HIV care and treatment clinic, but in a different room and by a different healthcare provider as for ART refills. Contraceptive refills were aligned with ART refills (i.e., 3 months oral pills for those who receive 3-month ART refills) and could be collected along with ART (although women could queue for certain methods). Two lower-level sites offered a one-stop shop approach, while in higher level hospitals, FP services were offered by a different provider in a different room than ART [30]. The Tanzania example lends credence to the fact that integrating FP into the SC-DSDM is possible (only that it is not happening as expected) and that WLHIV can ably access FP services through these models. However, what is less documented is how the integration of FP into the SC-DSDM affects FP service uptake, including whether or not FP and ART supplies have been synchronized. Future research is needed to assess the dynamics of FP uptake in a FP-HIV integrated framework to inform the scale-up of FP-HIV integration across programs and countries.

Our finding that FP information and counseling were provided to all clients at the surveyed health facilities demonstrates a very important step to ensure WLHIV have the information needed to reach their reproductive goals. Previous research shows that WLHIV who receive any FP counseling are significantly more likely to report current use of modern contraception than those who do not [31,32]. These findings support the need to ensure that FP counseling provided to WLHIV is of good quality and to enhance FP counseling when quality standards are not met. We, however, did not assess the quality of the FP information and counseling provided. Future research should focus on assessing quality and new tools to facilitate efficient provision of information and counseling in the context of staff shortages and numerous clients.

We found that most WLHIV accessed FP services through non-coordinated intra-facility referral to the MCH/FP clinic. This means that when WLHIV are referred from the ART clinic to the MCH/FP clinic, they have to line up afresh for FP services since there is no prior arrangement (coordination) between the two clinics to offer timely FP services to the women referred from the ART clinic. The absence of FP supplies at the ART clinic is a historical challenge that emanates from the fact that, traditionally, FP services at health facility level are provided through the MCH/FP clinic. We think that the traditional alignment of FP with MCH services makes HIV providers to feel less compelled to stock FP services as part of HIV care. While this may be speculative (since we did not specifically collect data on HIV service providers' attitudes towards providing FP services within the ART clinic), we believe that the long-standing availability of FP services at the MCH/FP clinic may call for a need to strengthen FP-HIV integration through coordinated intra-facility referral as an alternative model of integration, especially in situations where integration through the one-stop center may not be practically possible or achievable. Our findings already point to the fact that, in practice, there is more integration going on through collaboration than the one-stop center; thus, it may be helpful for the Ministry of Health and the donor community to explore the potential benefits of strengthening FP-HIV integration through referral to the MCH/FP clinic vis-à-vis promoting integration through the one-stop center that may be less feasible in some situations.

However, it is important to note that the women interviewed for this study indicated a preference for the one-stop center model of FP integration into the SC-DSDM. The acceptability of integration as one of the approaches necessary to improve health service delivery has been documented in other studies [33,34]. For instance, in a study conducted in Mayuge, Uganda, Mezei et al. [33] found that 99% of women and 100% of the surveyed community health workers felt that integrating community-based HIV testing with cervical cancer screening was acceptable. In this same study, women noted a preference for a combined service delivery model as it can save them time and eliminate additional trips to the health center, which have resource implications, in cases when services are not available on the same day. Women living with HIV face heightened vulnerability compared to their counterparts who do not live with HIV given that they have to manage a chronic condition for life. The importance of facilitating access to holistic health care that addresses their full needs has been highlighted as the HIV community moves towards more holistic care [35] and lends credence to the fact that offering FP methods and ART refills in the same place, at the same time, by the same or a different provider is not only acceptable but also *possible*. However, the one-stop shop model, usually dubbed as "combined service provision" in the case of integrating FP into immunization services [36], is likely harder to achieve than models that offer information/counseling plus referral given that it requires staff training and coordination such as the need to synchronize or integrate family planning and HIV commodity supply chains [37]. Studies also show that the implementation of the one-stop shop model can be hampered by patients' fear of stigma and discrimination (especially in community-based ART models), patients' and providers' low literacy levels on the DSD model and supply chain inconsistencies [29,38]. Thus, efforts geared at strengthening the provision of FP services through coordinated intra-facility referral may be the game-changer needed to improve access to FP services in most resource-constrained settings including Uganda.

While the Ministry of Health has already developed a National Strategy for Integration of Sexual Reproductive Health (SRH), HIV/AIDS, TB, Nutrition and Gender Based Violence (GBV) [39], we believe that the need for the development of implementation guidelines to support the integration of FP into the SC-DSDM remains essential. It is important to note that the above-mentioned SRH-HIV-GBV strategy examines the aspect of integration from the point of view of sexual and reproductive health rather than from the HIV care point of view, offering limited guidance on how and when to integrate FP into the SC-DSDM. To support the integration of FP into the DSDM in general and SC-DSDM in particular, it will be essential that the Ministry of Health develops specific DSDM-focused, FP-HIV integration implementation guidelines that define how, when and for whom the integration should happen in each health facility. The Ministry of Health should put in place mechanisms to ensure a constant supply of contraceptive supplies as well as provide for adequate office space to improve space available for the implementation of the one-stop center integration approach. Combined reporting tools that capture data on the use of both FP and ARV drug refills may need to be developed to avoid the need for completing two

different registers; one for the HIV care and the other for the FP supplies received. Finally, since we did not assess contraceptive uptake among WLHIV enrolled in the different SC-DSDM, we recommend further studies to assess differentials in FP uptake across the different SC-DSDM and document barriers/challenges and facilitators of contraceptive uptake under each DSD model within the FP-SC-DSDM integrated framework.

## Study limitations and strengths

This study was limited by the purposive sampling approach. Strictly interpreted, the results are only applicable to WLHIV who were interviewed for this study and not to the general population of WLHIV in the study regions in particular or Uganda in general. Nonetheless, the four study regions were chosen to represent a variety of local characteristics and we feel that the results will be useful to policymakers in designing guidelines for integrating FP into the SC-DSDM in other similar contexts. The other limitation is that this study relied on the self-reports of the clients and health providers regarding FP-HIV integration and the health facilities were categorized as integrating or not integrating based on this information. Because most WLHIV indicated that they obtained their FP services from the MCH/FP clinic, we categorized most health facilities as not integrating FP into the SC-DSDM without verifying that they actually obtained their FP methods from the MCH/FP clinics. If we did, this should have helped to inform our decision regarding whether or not the health facilities were actually integrating or not integrating FP into the SC-DSDM. However, we believe that this is a minor limitation. Given that we triangulated data collection methods and data sources (i.e., data were collected using in-depth interviews and key informant interviews with a diversity of participants including WLHIV, program implementers, district health managers, and policymakers), we believe that we made an accurate estimate of the FP-HIV integration status at the health facilities that we visited. Lastly, while we documented challenges/barriers inherent in implementing FP integration in the SC-DSDM, we relied on self-reports but did not try to quantify the magnitude of each barrier/challenge. For instance, we did not collect data on stock-outs of FP supplies but relied on self-reports during the provider interviews. This was because this study was conducted as a landscaping, qualitative study to inform future FP integration into the SC-DSDM rather than as a survey to obtain quantitative measures of each of the challenges/barriers noted.

These limitations notwithstanding, our study had a number of strengths. First and foremost, this study is one among the few studies that have assessed FP-HIV integration from the perspective of SC-DSDM. Thus, while studies on integration of HIV with other services, including family planning, have been previously conducted, these studies did not assess FP integration from the perspective of SC-DSDM. We, therefore, believe that the study findings can help to inform the design of FP-HIV integration interventions in the context of SC-DSDM, including facility and community-based models. Secondly, our study involved a total of 83 qualitative interviews (including in-depth interviews and key informant interviews) with different categories of participants, including clients, healthcare providers, district health managers, implementing partner representatives, and policy-makers, thereby improving the credibility of the findings through triangulation by data source and data collection methods.

## Conclusion

Our findings show that despite the potential benefits of integrating FP into the SC-DSDM, only four out of 18 health facilities offered FP services integrated into the SC-DSDM. Even in the few facilities that integrated FP into SC-DSDM, several challenges/barriers, including frequent stock-outs of short-term FP supplies, staff shortages, and lack of adequate office space continued to hamper effective integration. Unfortunately, while the integration of FP into the SC-DSDM is being spear-headed by the donor community, at the moment, there are no guidelines to inform the effective integration of FP into the SC-DSDM. These findings call for policy guidance from the Ministry of Health in integrating FP into the SC-DSDM in Uganda. This will not only help in ensuring that there is uniform implementation of FP integration into the SC-DSDM but also provide guidance on how to address the barriers that currently inhibit the effective integration of FP into SC-DSDM in Uganda.

## Supporting information

**S1 File. S1**. In-depth interview guide for women living with HIV. **S2**. Interview guide for health facility staff. **S3**. Key informant interview guide for program managers. **S4**. Key informant interview guide for policymakers.
(ZIP)

## Acknowledgments

We are greatly indebted to Rawlance Ndejjo, Sarah Nabukeera, and Noel Namuhani (Research Officers working with the Research for Scalable Solutions [R4S] Project) who supervised the data collection process and participated in the analysis of data. We thank the participants at all levels (policymakers at the Ministry of Health, implementing partner staff, district health officers, and women living with HIV, enrolled in the different DSD models) for agreeing to participate in the study. Finally, we are grateful for the support that we received from the Ugandan Ministry of Health, FHI360 Uganda office and the USAID/Uganda office during the conduct of this study.

## Author contributions

**Conceptualization:** Joseph K. B. Matovu, Fredrick Makumbi, Elena Lebetkin, Rhobbinah Ssempebwa, Patrick Komakech, Dieudonne Bidashimwa, Rhoda K. Wanyenze.

**Data curation:** Martha Akulume, Maria Carrasco.

**Formal analysis:** Joseph K. B. Matovu, Martha Akulume.

**Funding acquisition:** Fredrick Makumbi, Elena Lebetkin, Rhoda K. Wanyenze.

**Investigation:** Joseph K. B. Matovu, Martha Akulume, Fredrick Makumbi, Rhobbinah Ssempebwa, Patrick Komakech, Dieudonne Bidashimwa.

**Methodology:** Joseph K. B. Matovu, Martha Akulume, Elena Lebetkin, Dieudonne Bidashimwa, Rhoda K. Wanyenze.

**Supervision:** Joseph K. B. Matovu, Martha Akulume.

**Validation:** Joseph K. B. Matovu, Martha Akulume, Elena Lebetkin, Rhobbinah Ssempebwa, Patrick Komakech, Dieudonne Bidashimwa, Maria Carrasco, Rhoda K. Wanyenze.

**Visualization:** Rhobbinah Ssempebwa, Dieudonne Bidashimwa, Maria Carrasco, Rhoda K. Wanyenze.

**Writing – original draft:** Joseph K. B. Matovu, Martha Akulume, Fredrick Makumbi, Elena Lebetkin, Rhobbinah Ssempebwa, Patrick Komakech, Dieudonne Bidashimwa, Maria Carrasco, Rhoda K. Wanyenze.

**Writing – review & editing:** Joseph K. B. Matovu, Fredrick Makumbi, Rhoda K. Wanyenze.

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
