## [Decision Letter · Decision Letter 0]

17 Mar 2025

PONE-D-24-53280Stakeholders’ Perspectives on the Status of Family Planning Integration into Differentiated Antiretroviral Therapy Service Delivery Models in Uganda: A Qualitative AssessmentPLOS ONE

Dear Dr. Matovu,

Thank you for submitting your manuscript to PLOS ONE. After careful consideration, we feel that it has merit but does not fully meet PLOS ONE’s publication criteria as it currently stands. Therefore, we invite you to submit a revised version of the manuscript that addresses the points raised during the review process.

We look forward to receiving your revised manuscript.

Kind regards,

Tinei Shamu

Academic Editor

PLOS ONE

Journal Requirements:

Additional Editor Comments:

Congratulations on putting together this manuscript. Please attend to the comments from the reviewers.

Reviewers' comments:

Reviewer's Responses to Questions

**Comments to the Author**

1. Is the manuscript technically sound, and do the data support the conclusions?

Reviewer #1: Yes

Reviewer #2: Yes

2. Has the statistical analysis been performed appropriately and rigorously? 

Reviewer #1: N/A

Reviewer #2: N/A

3. Have the authors made all data underlying the findings in their manuscript fully available?

Reviewer #1: No

Reviewer #2: No

4. Is the manuscript presented in an intelligible fashion and written in standard English?

Reviewer #1: Yes

Reviewer #2: Yes

5. Review Comments to the Author

Reviewer #1: This is a very comprehensive and detailed paper. It was easy to follow and the writing was clear. It was also an enjoyable paper to read as it made a strong case for integrating family planning into DSD models. In addition to capturing women's own experiences, I got a really deep understanding of the health facility-level challenges. Even though staff shortages etc are commonly reported challenges in studies such as this one, the focus on integration challenges specifically was very illuminating. Overall, the authors have done an excellent job. My one major concern, however, is that the paper is way too long. It would have an even greater impact and grab the reader more effectively if it was about 25% shorter. Every section is described in great detail, which can be a little too much. Consider cutting down on the methods, which are currently 6 pages long, to perhaps half the length. The discussion and/or the introduction could also be made more succinct. This is my only concern, but I will defer to the editorial policy of the journal in terms of the length of articles.

Reviewer #2: Thank you for the opportunity to review the manuscript “Stakeholders’ Perspectives on the Status of Family Planning Integration into Differentiated Antiretroviral Therapy Service Delivery Models in Uganda: A Qualitative Assessment”. The authors offer a well-written account of perspectives from women living with HIV, service providers, health managers, implementing partner representatives and policymakers on FP-HIV integration from the perspective of self-care-oriented DSDM in Uganda, which gives this work a novel lens and point of view. The following suggestions might help impove the manuscript:

1. Line 229. I think you meant to say you sought consent from all the clients before their participation or that participants gave their consent instead of “All clients consented before their interviews”.

2. It would be helpful to have a clearer picture of the data analysis. Could you please provide more detail on the data extraction, coding, and categorisation or theme development process? Specifically, it's essential to know how many people were involved and their roles. Also, clarifying the analytical framework that guided the work would be beneficial. This would strengthen the manuscript's credibility and help address any potential concerns about bias.

3. Line 329-336 – what does having FP integrated into the SC-DSDM at a given health facility mean? Did the service providers and the clients have a shared understanding of the concept?

4. It may be an issue of preference, but I suggest presenting the results first and discussing them later. For example, the following statements could wait for the discussion section.

a. Line 389-90 – In general, evidence shows that the community-based DSD models have the lowest number of enrolled clients when compared to the other DSD models [25].

b. Line 454 - Staff shortages have been previously reported as key barriers in the implementation of DSDM [26]

c. Line 466-7 - Effective implementation of DSDM relies on the constant availability of ART drugs across the different ART models of care [25].

5. Line 395-96 – Is there evidence from the data to support this claim – “Our findings show that FP-HIV integration into the DSDM was not comprehensively discussed during the design of the guidelines….” Or it’s the interpretation of the authors?

6. Line 617-618 – It might help to qualify this statement and add a reference(s) – “…given the efforts put into supporting FP-HIV integration in SC-DSDM by the donor community in Uganda.”

7. Could you share the final version of the Health Facility Staff Interview Guide? The current version has changes that have not been accepted or finalised.

6. PLOS authors have the option to publish the peer review history of their article (what does this mean? ). If published, this will include your full peer review and any attached files.

**Do you want your identity to be public for this peer review?** For information about this choice, including consent withdrawal, please see our Privacy Policy .

Reviewer #1: No

Reviewer #2: **Yes: ** Farirai Mutenherwa

---

## [Author Response · Author response to Decision Letter 1]

31 Mar 2025

We have provided a response to the comments in our 'Response to Reviewers' document.

---

## [Decision Letter · Decision Letter 1]

29 Apr 2025

Stakeholders’ Perspectives on the Status of Family Planning Integration into Differentiated Antiretroviral Therapy Service Delivery Models in Uganda: A Qualitative Assessment

PONE-D-24-53280R1

Dear Dr. Matovu,

We’re pleased to inform you that your manuscript has been judged scientifically suitable for publication and will be formally accepted for publication once it meets all outstanding technical requirements.

Kind regards,

Tinei Shamu

Academic Editor

PLOS ONE

Additional Editor Comments (optional):

Reviewers' comments:

Reviewer's Responses to Questions

**Comments to the Author**

1. If the authors have adequately addressed your comments raised in a previous round of review and you feel that this manuscript is now acceptable for publication, you may indicate that here to bypass the “Comments to the Author” section, enter your conflict of interest statement in the “Confidential to Editor” section, and submit your "Accept" recommendation.

Reviewer #1: All comments have been addressed

Reviewer #2: All comments have been addressed

2. Is the manuscript technically sound, and do the data support the conclusions?

Reviewer #1: Yes

Reviewer #2: Yes

3. Has the statistical analysis been performed appropriately and rigorously? 

Reviewer #1: N/A

Reviewer #2: N/A

4. Have the authors made all data underlying the findings in their manuscript fully available?

Reviewer #1: Yes

Reviewer #2: Yes

5. Is the manuscript presented in an intelligible fashion and written in standard English?

Reviewer #1: Yes

Reviewer #2: Yes

6. Review Comments to the Author

Reviewer #1: All my comments have been addressed satisfactorily. I have no further comments to add. Good luck with publication.

Reviewer #2: (No Response)

7. PLOS authors have the option to publish the peer review history of their article (what does this mean? ). If published, this will include your full peer review and any attached files.

**Do you want your identity to be public for this peer review?** For information about this choice, including consent withdrawal, please see our Privacy Policy .

Reviewer #1: No

Reviewer #2: **Yes: ** Farirai Mutenherwa

---

## [Editor Report · Acceptance letter]

PONE-D-24-53280R1

PLOS ONE

Dear Dr. Matovu,

I'm pleased to inform you that your manuscript has been deemed suitable for publication in PLOS ONE. Congratulations! Your manuscript is now being handed over to our production team.

Kind regards,

on behalf of

Dr. Tinei Shamu

Academic Editor

PLOS ONE